# EasyRef: Omni-Generalized Group Image Reference for Diffusion Models via Multimodal LLM

**Zhuofan Zong** [1 2]  **Dongzhi Jiang** [1]  **Bingqi Ma** [2]  **Guanglu Song** [2]
**Hao Shao** [1]  **Dazhong Shen** [3]  **Yu Liu** [2]  **Hongsheng Li** [1 4 5]

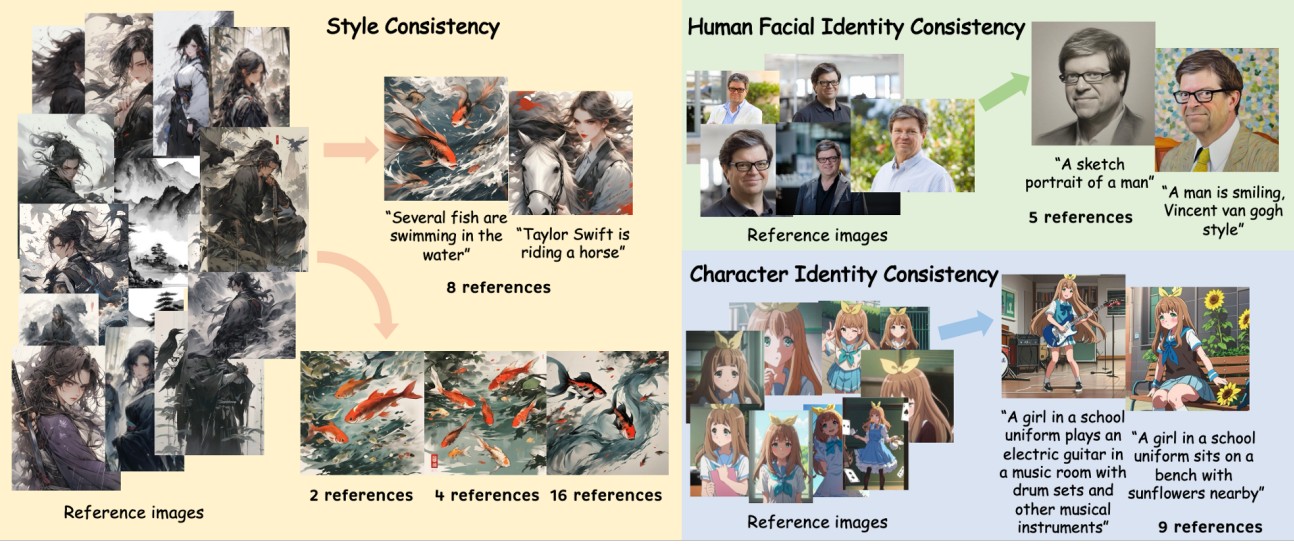

*Figure 1.* **EasyRef** adopts a single multimodal LLM to capture consistent visual elements such as style, character identity, and human facial identity across various reference images and generates customization results with a diffusion model.

## Abstract

Significant achievements in personalization of diffusion models have been witnessed. Conventional tuning-free methods mostly encode multiple reference images by averaging or concatenating their image embeddings as the injection condition, but such an image-independent operation cannot perform interaction among images to capture consistent visual elements within multiple references. Although tuning-based approaches can effectively extract consistent elements within multiple images through the training process, it necessitates test-time finetuning for each distinct image group. This paper introduces EasyRef, a plug-and-play

adaption method that empowers diffusion models to condition consistent visual elements (e.g., style and human facial identity, etc.) across multiple reference images under instruction controls. To effectively exploit consistent visual elements within multiple images, we leverage the multi-image comprehension and instruction-following capabilities of the multimodal large language model (MLLM), prompting it to capture consistent visual elements based on the instruction. Besides, injecting the MLLM's representations into the diffusion process through adapters can easily generalize to unseen domains. To mitigate computational costs and enhance fine-grained detail preservation, we introduce an efficient reference aggregation strategy and a progressive training scheme. Finally, we introduce MRBench, a new multi-reference image generation benchmark. Experimental results demonstrate EasyRef surpasses both tuning-free and tuning-based methods, achieving superior aesthetic quality and robust zero-shot generalization across diverse domains.

[1]CUHK MMLab [2]SenseTime Research [3]Nanjing University of Aeronautics and Astronautics [4]Shanghai AI Laboratory [5]CPII under InnoHK. Correspondence to: Hongsheng Li <hsli@ee.cuhk.edu.hk>.

*Proceedings of the 42nd International Conference on Machine Learning*, Vancouver, Canada. PMLR 267, 2025. Copyright 2025 by the author(s).

# 1. Introduction

Significant achievements in diffusion models (Rombach et al., 2022; Podell et al., 2023; Betker et al., 2023; Esser et al., 2024; Ramesh et al., 2022; 2021; Saharia et al., 2022) have been witnessed because of their remarkable abilities to create visually stunning images. To improve the precision and controllability of diffusion models, researchers have been exploring personalized generation conditioned on a small number of reference images, categorized into test-time tuning-based methods (Hu et al., 2021; Ruiz et al., 2023; Gal et al., 2022) and tuning-free methods (Ye et al., 2023; Wang et al., 2024e;a; Zhang et al., 2023; Li et al., 2025).

Despite the promise of tuning-free methods, they have several limitations. First, encoders tailored for specific reference elements, such as style or facial identity, often rely on complex, task-specific architectures and specialized training processes. Second, most methods (Ye et al., 2023; Wang et al., 2024a; Qi et al., 2024) are limited to training with a single reference image and fail to fully encode consistent visual representations from multiple references. For instance, as illustrated in Figure 2, IP-Adapter (Ye et al., 2023) with average embeddings encounters two issues: (1) Attribute confusion arises when reference images have overlapping subjects (e.g., a dog partially or fully covering a chair), causing the averaged features to incorrectly blend attributes (e.g., the dog acquiring the chair's color and vice versa). (2) Subject hallucination occurs when positional discrepancies between reference subjects (e.g., a dog positioned in front of a chair versus seated on it) mislead the fusion method, resulting in the diffusion model erroneously generating an extra dog-shaped object on the chair. Although tuning-based methods can extract consistent elements within multiple images through the training process, it necessitates finetuning for each distinct image group.

This paper introduces EasyRef, a plug-and-play adaption method that empowers diffusion models to condition consistency (e.g., style, character identity, human facial identity, etc.) across multiple reference images under instruction controls. Conventional methods encode consistent elements across reference images through averaging or concatenation (Shi et al., 2024), but these image-independent operations fail to capture desired visual elements through effective image interaction under explicit controls. To alleviate this issue, we leverage the multi-image comprehension and instruction-following capabilities of the multimodal large language model (MLLM) (Liu et al., 2024b;a; Li et al., 2023; Chen et al., 2024), prompting it to capture consistent visual elements based on the instruction. Besides, injecting the MLLM's representations into the diffusion process through adapters can easily generalize to unseen domains, mining the consistent visual elements within unseen data. EasyRef also inherits the MLLM's ability to process arbitrary number

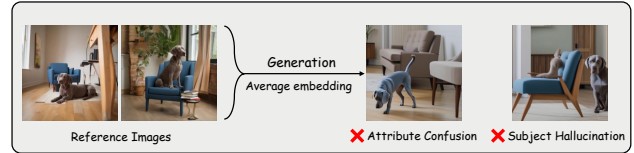

*Figure 2.* Attribute confusion and subject hallucination issues of the embedding averaging operation.

of reference images with arbitrary aspect ratios. To mitigate the computational demands imposed by the long context of multi-image inputs, we propose querying the MLLM and aggregating reference representations through reference tokens in the final layer of the MLLM architecture. Additionally, to address the limitations of MLLMs in capturing fine-grained visual details, we employ a progressive training strategy to enhance the MLLM's capacity for fine-grained detail and human facial identity preservation. Unlike previous methods that rely on sophisticated and task-specific feature encoders (Radford et al., 2021; Oquab et al., 2023; Deng et al., 2019), we demonstrate that the single MLLM in EasyRef can effectively extract diverse consistent reference representations, including style, character identity, object identity, and human facial identity, from an arbitrary group of reference images under instruction controls, while also exhibiting strong generalization ability. Finally, we introduce a multi-reference consistent generation benchmark (MRBench) for multi-reference consistent image generation to evaluate our work and guide future research. Compared to prevalent tuning-free IP-Adapter and tuning-based Low-Rank Adaptation (LoRA), EasyRef achieves superior aesthetic quality and reference consistency performances across diverse domains and demonstrates robust generalization.

In summary, our contributions are threefold: (1) We introduce EasyRef, a plug-and-play method that empowers diffusion models to condition various consistency across multiple reference images under instruction controls. (2) We propose an efficient reference aggregation strategy and a progressive training scheme to mitigate computational costs and enhance the MLLM's fine-grained perceptual abilities. (3) A novel MRBench is proposed for evaluating diffusion models in multi-reference consistent generation scenarios.

# 2. EasyRef

## 2.1. Methodology

As illustrated in Figure 3, EasyRef comprises four key components: (1) a pretrained diffusion model for conditional image generation, (2) a pretrained multimodal large language model (MLLM) for encoding a set of reference images and the instruction, (3) a condition projector that maps the representations from the MLLM into the latent space of diffusion model, and (4) trainable adapters for integrating

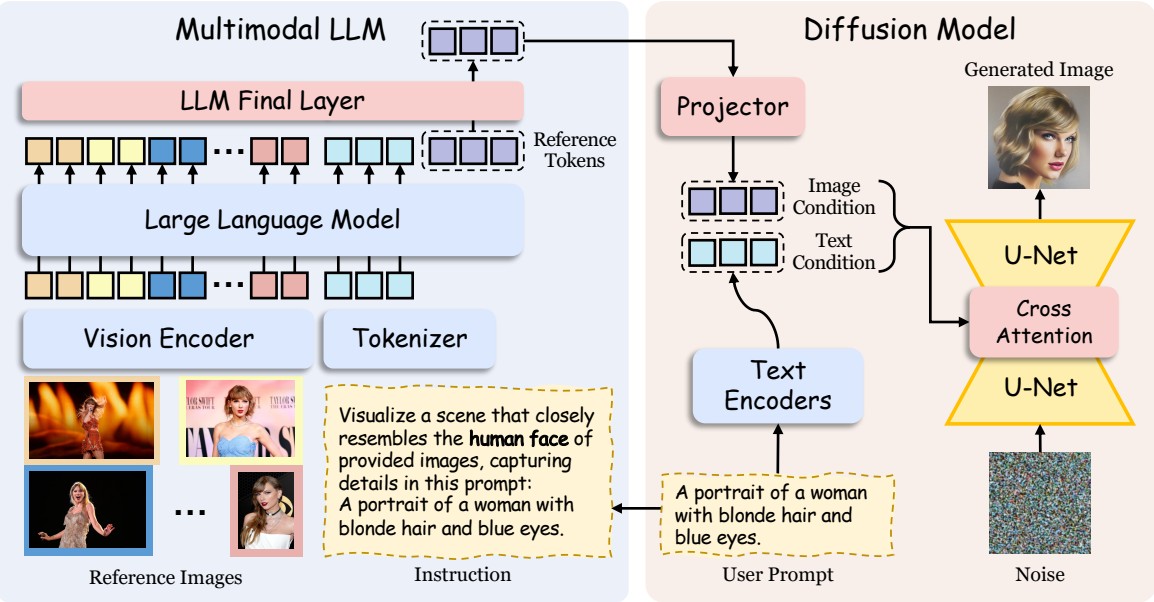

*Figure 3.* **Overview of EasyRef with Stable Diffusion XL.** EasyRef extracts consistent visual eliments from multiple reference images and the text prompt via a MLLM, injecting the condition representations into the diffusion model through cross-attention layers. We only plot 1 cross-attention layer for simplicity.

image conditioning embedding into the diffusion process.

**Reference Representation Encoding.** Existing mainstream approaches (Ye et al., 2023; Qi et al., 2024; Wang et al., 2024a;e) mostly average or concatenate the image embeddings of all reference images as the condition. These image-independent operations cannot effectively capture consistent visual elements among reference images. It also fails to encode reference information under explicit instruction control and causes the spatial misalignment issue as presented in Figure 2. To alleviate this issue, we propose to leverage the multi-image comprehension and instruction-following capabilities of the MLLM to encode multi-reference inputs based on the instruction. We adopt the state-of-the-art Qwen2-VL-2B as our MLLM in this work. The MLLM consists of a $l$-layer large language model (LLM) and a vision encoder capable of handling images with arbitrary resolutions. The input image is initially converted into visual tokens with the vision encoder. Then we employ an instruction and integrate all images into the instruction, which explicitly encourages the MLLM to focus on desired consistent visual elements within the reference images. These multimodal input tokens are subsequently processed by the LLM.

**Efficient Reference Aggregation.** Increasing the number of reference images inevitably raises the number of visual tokens in the LLM. This extended context length substantially elevates the computational cost for the diffusion model. We propose to encapsulate the reference representations into $N$ learnable reference tokens $\mathbf{F}_{\text{ref}} \in \mathbb{R}^{N \times D}$ in the LLM

to achieve efficient inference. However, all parameters of LLM must be trained to interpret these newly added tokens. To enhance training efficiency, we append $\mathbf{F}_{\text{ref}}$ to the context sequence $\mathbf{F}_{l-1}$ at the final layer of LLM, keeping all previous LLM layers frozen during alignment pretraining:

$$\mathbf{F}'_l = \text{Concat}(\mathbf{F}_{l-1}, \mathbf{F}_{\text{ref}}) \tag{1}$$

Then we employ bi-directional self-attention to facilitate the propagation of representations across the reference images in the final layer, followed by a multi-layer perception network (MLP):

$$\mathbf{F}''_l = \text{MLP}(\text{Bi-Attention}(\mathbf{F}'_l)), \tag{2}$$

where we omit the residual addition operations for simplicity. Next, we split $\mathbf{F}''_l$ into the updated representations $\mathbf{F}_l$ and the encapsulated reference tokens $\mathbf{F}'_{\text{ref}}$:

$$\mathbf{F}_l, \mathbf{F}'_{\text{ref}} = \text{Split}(\mathbf{F}''_l). \tag{3}$$

Finally, we project $\mathbf{F}'_{\text{ref}}$ through a trainable MLP condition projector to obtain the final conditioning vector $\mathbf{c}_i$:

$$\mathbf{c}_i = \text{MLP}(\mathbf{F}'_{\text{ref}}), \tag{4}$$

**Reference Representation Injection.** The text conditions are injected into the pretrained diffusion model through cross-attention layers. Following IP-Adapter, we introduce a new cross-attention layer into each cross-attention layer of the U-Net. Given the latent features $\mathbf{X}$, text conditions

*Table 1.* Comparison of EasyRef against other counterparts.

| Method | Consistency encoding | Multiple references | Instruction | Tuning-free |
|---|---|---|---|---|
| LoRA (Hu et al., 2021) | ✔ | ✗ | ✗ | ✗ |
| IP-Adapter (Ye et al., 2023) | ✗ | ✗ | ✗ | ✔ |
| Kosmos-G (Pan et al., 2023) | ✗ | ✔ | ✔ | ✔ |
| MoMA (Song et al., 2025) | ✗ | ✗ | ✔ | ✔ |
| EasyRef | ✔ | ✔ | ✔ | ✔ |

$c_t$, and image conditions $c_i$, the injected features $\hat{X}$ are computed by the cross-attention layer as follows:

$$\hat{X} = \text{Softmax}\left(\frac{QK^T}{\sqrt{d}}\right)V + \text{Softmax}\left(\frac{Q\hat{K}^T}{\sqrt{d}}\right)\hat{V}, \quad (5)$$

where $\hat{K} = c_i\hat{W}_k$ and $\hat{V} = c_i\hat{W}_v$. Both $\hat{W}_k$ and $\hat{W}_v$ are newly added trainable parameters.

## 2.2. Progressive Training Scheme

**Alignment Pretraining.** To facilitate the adaption of MLLM's visual signals to the diffusion model, we construct a large-scale dataset containing 13M high-quality image-text pairs, including LAION-5B (Schuhmann et al., 2022) and other internal datasets for the alignment pretraining. During the pretraining phase, we only optimize the final layer and reference tokens of the MLLM along with the newly added adapters and condition projector while preserving the capabilities of the initial MLLM and diffusion model. The model is pretrained for 300k iterations. We center crop $1024 \times 1024$ pixels of the input image.

**Single-reference Finetuning.** Following alignment pretraining, the MLLM is trainable and subjected to single-reference finetuning. Specifically, we unfreeze the vision encoder and all layers of the MLLM to enhance its capacity for fine-grained visual perception at the second stage. We additionally incorporate trainable Low-Rank Adaption (LoRA) layers to attention layers of the frozen U-Net. Building upon the aforementioned pretraining dataset, we augment the training data with 4M real-world human images from LAION-5B, utilizing cropped face regions as conditioning inputs for better human facial identity preservation. Thanks to the efficient aggregation design and alignment pretraining, we only train the model for 80k iterations.

**Multi-reference Finetuning.** The third stage enables the MLLM to accurately comprehend the consistent visual elements across multiple image references under instruction controls. Training is performed on a curated dataset comprising image groups, where each group contains multiple images of the same topic (*e.g.*, a Tesla Cybertruck, *etc.*) with varying aspect ratios. During training, one image from each group is randomly selected as the optimization target, while the remaining ones serve as the conditioning inputs. Data augmentation, including random shuffling and truncation,

is applied to the conditioning images. We keep the original aspect ratio for each target image.

**Training Supervision.** We use the same training objective as the original stable diffusion model:

$$\mathcal{L} = \mathbb{E}_{x_0, \epsilon, c_t, c_i, t} \|\epsilon - \epsilon_\theta(x_t, c_t, c_i, t)\|^2, \quad (6)$$

where $c_t$ and $c_i$ denote the text condition and image condition, respectively.

## 2.3. Discussion

**Comparison with Other Methods.** We compare EasyRef with LoRA, IP-Adapter (Ye et al., 2023), and other image personalization methods that adopt MLLMs in Table 1. First, the task differs significantly. Compared to Kosmos-G (Pan et al., 2023) and MoMA (Song et al., 2025), both EasyRef and LoRA possess the capability to encode consistent visual elements, such as style, subject, and human face, within a group of reference images. While Kosmos-G primarily focuses on the combination of different elements from multiple images, MoMA utilizes a MLLM to extract the subject feature of the single reference for subject-driven generation. Second, the reference aggregation paradigm is different. Kosmos-G employs an elaborate encoder-decoder AlignerNet, while MoMA utilizes learnable tokens and subject-aware masked attention to bridge the MLLM to the diffusion U-Net. In contrast, we demonstrate that simply using reference tokens in the final layer of the MLLM can provide sufficient reference information and efficiently inject conditions into the U-Net. Additionally, we propose a progressive training scheme for the MLLM to enable the extraction of fine-grained details (e.g., human facial details).

## 3. Multi-Reference Generation Benchmark

**Data Source.** Our data source encompasses two parts: (1) We collect images from several large-scale publicly available datasets, including LAION-2B (Schuhmann et al., 2022), COYO-700M (Byeon et al., 2022), and DataComp-1B (Gadre et al., 2024). (2) We also constructed a tag list that includes celebrity names, character names, styles, and subjects, and collected filtered images from diverse sources based on this list. Images sharing the same tag are set into the same group. Subsequently, these images were used to train LoRA models for each group and we incorporated high-quality images generated by these models into our training data.

**Dataset Construction.** To generate aligned text captions of the images, synthetic captions generated by Qwen2-VL-7B using the instruction "*Give a brief, concise and precise caption for this image.*" were adopted for each sample. To achieve controllable reference encoding, we annotate the

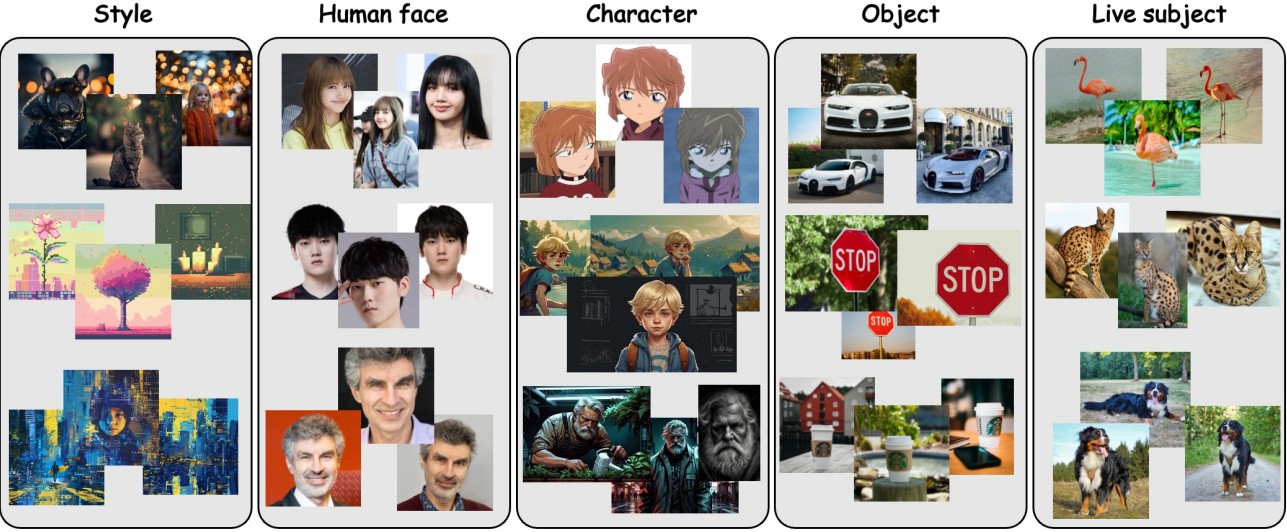

Figure 4. Example images for each group in our proposed MRBench. We only present 3 images for each group.

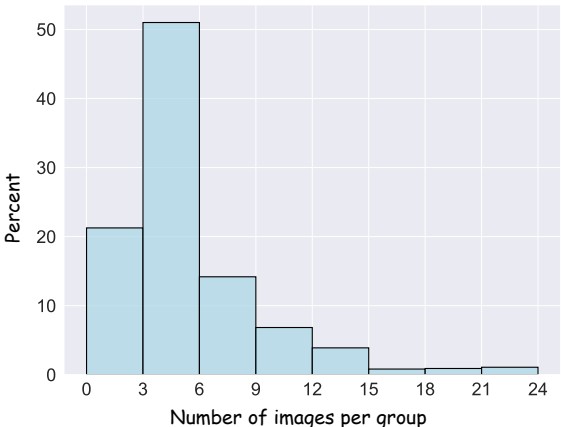

Figure 5. Distribution of our curated dataset.

consistent visual elements of each group using Qwen2-VL-72B. Then we explicitly ask the MLLM to capture desired references by incorporating the annotated consistent elements into instructions during training. The resulting dataset comprises 3,918,984 images organized into 994,275 groups. We set the maximum and minimum group sizes as 16 and 2, respectively, ensuring a balanced group size distribution within the dataset. Figure 5 illustrates the group distribution of our curated dataset.

**Data Filtering.** We also employ a series of efforts for data cleaning. Initially, images with low resolution, poor aesthetic scores, or extreme aspect ratios were excluded. Then we filter out image-text pairs with low CLIP image-text similarity scores using CLIP ViT-L/14 (Radford et al., 2021). Since the aforementioned filtering methods may not effectively identify special patterns such as image collages, high-quality images with dense text, watermarks, etc., we

manually annotate a subset of the collected samples. The valid images are labeled as positive, while the images to be filtered are labeled as negative. We then train a CLIP-based binary classifier to filter these negative instances. Finally, we perform deduplication to eliminate redundant images.

**Data Clustering.** We construct facial identity groups and character identity groups based on a predefined tag list. For images containing human faces or characters, we use Co-DETR (Zong et al., 2023b) to detect face bounding boxes and character body boxes. Face embeddings and character embeddings are then extracted using ArcFace (Deng et al., 2019) and CLIP, respectively. For each facial or character identity group, we calculate the cosine similarity between the embeddings of the current group and those of newly collected samples. The similarity scores for each new sample are summed, and the group with the highest score is assigned to the sample. Samples with low similarity scores are discarded to ensure relevance. Groups for style and subject (e.g., common objects and animals) are constructed in a similar manner. We use CLIP to extract style embeddings and DINOv2 (Oquab et al., 2023) to extract subject embeddings. Unlike the previous strategy, a new group is created if a collected sample does not match any existing group. Besides, we do not only assign the group with highest score to the sample and a sample can be assigned to multiple groups. For instance, an image of a Bernese Mountain Dog can belong to both the Bernese Mountain Dog Group and the Dog Group. We find that subjects belonging to the same category but not the same instance can be grouped together. However, this does not significantly impact subject-driven performance. Multi-image subject-driven generation requires that the main subjects across multiple reference images represent the same instance, and the generated outputs should contain

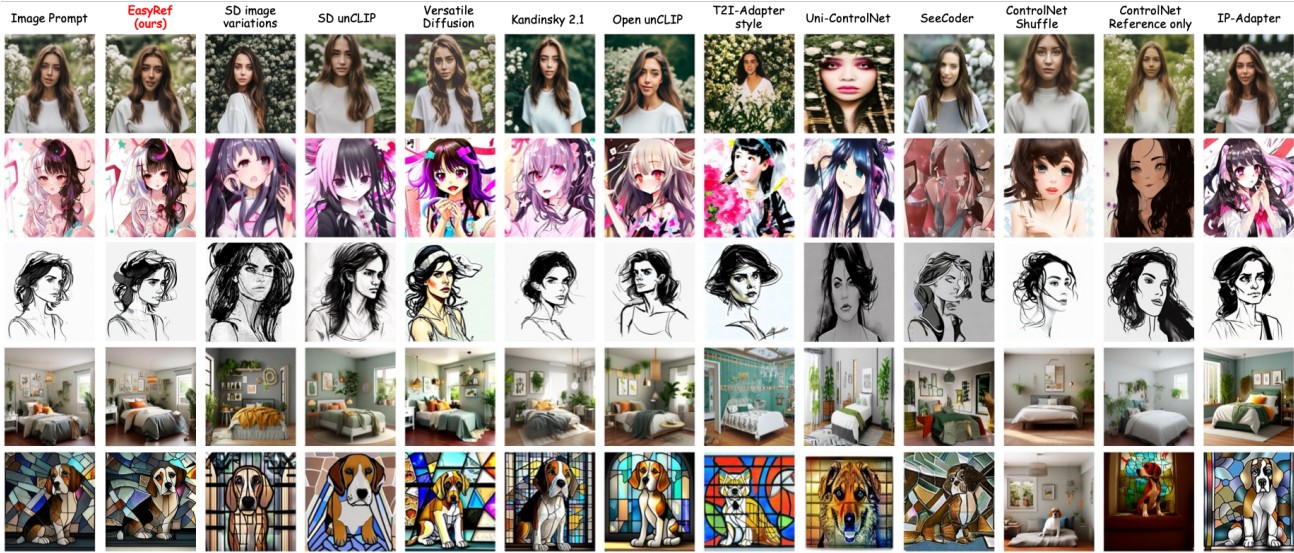

Figure 6. Comparisons of EasyRef with other counterparts in various single-image reference scenarios. The same image prompts as in (Ye et al., 2023) are used for clear comparisons.

the same instance. When multiple reference images depict subjects from the same category but not the same instance, using a target image with a subject from the same category is still a reasonable training approach.

**Benchmark Splits.** The collected image-text pairs are divided into the training dataset, the held-in evaluation set, and the held-out evaluation set. We first sample 50 image groups to construct the held-out evaluation set to evaluate the model performance on unseen data. The number of images in each set varies. There are a total of 300 images in the held-out evaluation set. To avoid data contamination, we use the CLIP image embeddings of the held-out images to retrieve and filter out similar images from the remaining data. To compare our method with multi-reference generation approaches that require finetuning (*e.g.*, LoRA), we randomly selected 300 groups from the remaining 994,215 groups to form a test set of 1434 samples. Unlike the held-out set, only a randomly selected image serves as the target image in each group. All reference images of the held-in split and other 993,915 groups construct the training set. To improve the aesthetic quality of generated images, only images with aesthetic scores higher than 5.5 can be used as the training target images. There are 2,117,435 valid training target images in the training set, with an average of 2.1 images per group.

**Benchmark Statistics.** We categorize the consistent visual elements of each group in the benchmark into five categories: style, human faces, characters, objects, and live subjects/pets. Specifically, the held-out dataset includes 25 style groups, 10 human face groups, 5 character groups, 5 object groups, and 5 groups of live subjects.

**Dataset Samples.** As shown in Figure 4, we provide some samples of MRBench. To ensure the benchmark's high quality, we manually collect, verify, and integrate selected images into the MRBench. The synthetic captions are generated using Qwen2-VL-7B, and detailed descriptions are removed from the captions to prevent content leakage.

**Evaluation Protocol.** For each group of evaluation data, each image can be chosen as the target image and others are regarded as the reference images. When evaluating the held-in and held-out sets, we use the reference images and caption of the target image to generate two images for each group. Then we employ conventional metrics, including CLIP-I, CLIP-T, and DINO-I, to measure the alignment between generated images and the corresponding target images or prompts. We mainly consider CLIP-I and DINO-I for the image-image alignment, which computes the similarities of image embeddings from CLIP ViT-L/14 and DINOv2-Small (Oquab et al., 2023). For the image-text alignment, we adopt the CLIPScore (Hessel et al., 2021).

**Comparison with Other Benchmarks.** The Dream-Bench (Ruiz et al., 2023) benchmark is a pioneering dataset for evaluating multi-reference generation. However, there are several key differences between it and MRBench. First, MRBench covers a more diverse range of categories, spanning five distinct types, while the DreamBench focuses solely on subject-driven generation for objects and live subjects or pets. Second, MRBench comprises 300 meticulously curated images, compared to only 158 images in the DreamBench. Third, the evaluation protocols differ significantly. In MRBench, each image, along with its caption, can be selected as the target, while the remaining images in the

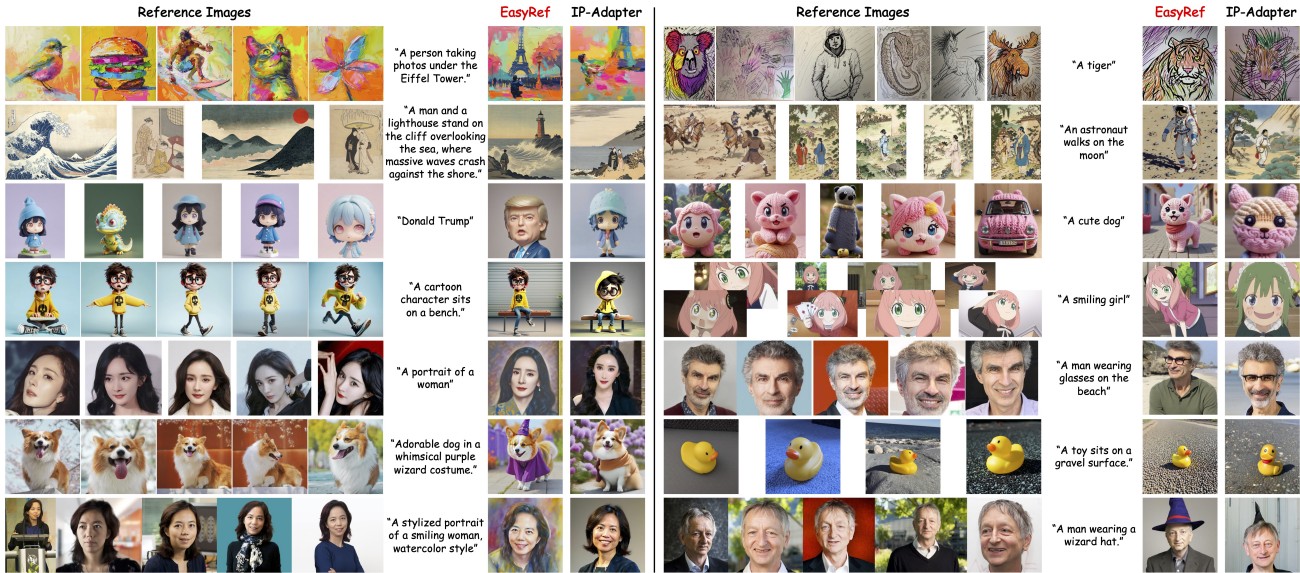

*Figure 7.* Visualization of generated samples with various multi-reference inputs. Best viewed by zooming in. Note that the base diffusion model cannot generate these reference images of celebrities and characters with text prompts.

group serve as the conditioning inputs. Visual consistency is then evaluated between the generated images and the single target image. In contrast, the DreamBench treats all images as references and evaluates the average similarity between the generated and reference images.

## 4. Experiments

### 4.1. Implementation Details

**Training.** We build our EasyRef with the established Stable Diffusion XL (Podell et al., 2023) model, utilizing the state-of-the-art Qwen2-VL-2B (Wang et al., 2024d) as the MLLM. The resolution of an input image with arbitrary aspect ratio processed by the MLLM can not exceed $336 \times 336$. We introduce 64 reference tokens in the MLLM. We also employ a drop probability of 0.1 for both text and image prompts independently, and a joint drop probability of 0.1 for simultaneous removal of both modalities. We simply treat a square black image as the empty image condition if the image condition is dropped. For the implementation of LoRA comparison, we fine-tuned the model using the reference images and employed a LoRA rank of 32.

**Evaluation.** During inference, we leverage a DDIM (Song et al., 2020) sampler with 30 steps and a guidance scale (Ho & Salimans, 2022) of 7.5. As the original IP-Adapter does not support multi-image references, we employed the average of the CLIP embeddings as the conditioning input. The reference images presented in our visualizations are excluded from the training data. We present more experimental results in the Appendix A.2.

*Table 2.* Performance comparisons on COCO validation set. Methods with * use CLIP embeddings and tend to achieve higher scores of CLIP-based metrics due to its preference.

| Method | CLIP-I ↑ | CLIP-T ↑ | DINO-I ↑ |
|---|---|---|---|
| *Training from scratch* | | | |
| Open unCLIP (Ramesh et al., 2022) | 0.858 | 0.608 | - |
| Kandinsky-2-1 (Arseniy et al., 2023) | 0.855 | 0.599 | - |
| Versatile Diffusion (Xu et al., 2023) | 0.830 | 0.587 | - |
| *Finetuning* | | | |
| SD Image Variations | 0.760 | 0.548 | - |
| SD unCLIP | 0.810 | 0.584 | - |
| *Adapters* | | | |
| Uni-ControlNet (Zhao et al., 2024) (Global Control) | 0.736 | 0.506 | - |
| T2I-Adapter (Mou et al., 2024) (Style) | 0.648 | 0.485 | - |
| ControlNet Shuffle (Zhang et al., 2023) | 0.616 | 0.421 | - |
| IP-Adapter* (Ye et al., 2023) | 0.828 | 0.588 | - |
| IP-Adapter-SDXL* (Ye et al., 2023) | 0.836 | 0.617 | 0.650 |
| EasyRef | **0.876** | **0.621** | **0.873** |

### 4.2. Quantitative and Qualitative Results

**Single-image Reference.** We quantitatively compare our method with other counterparts in single-reference scenarios using the COCO 2017 validation dataset (Lin et al., 2014), which comprises 5000 image-text pairs. We use the checkpoint trained by single-reference finetuning. As shown in Table 2, EasyRef consistently outperforms other methods in both CLIP-T and DINO-I metrics, demonstrating superior alignment performance. For instance, our model significantly surpasses the IP-Adapter-SDXL by 0.223 DINO-I score. Note that IP-Adapter utilizes CLIP image embeddings for conditioning, its generated images may exhibit a bias towards CLIP's preference, potentially increasing scores when evaluated using CLIP-based metrics. We further conduct qualitative comparisons using some reference

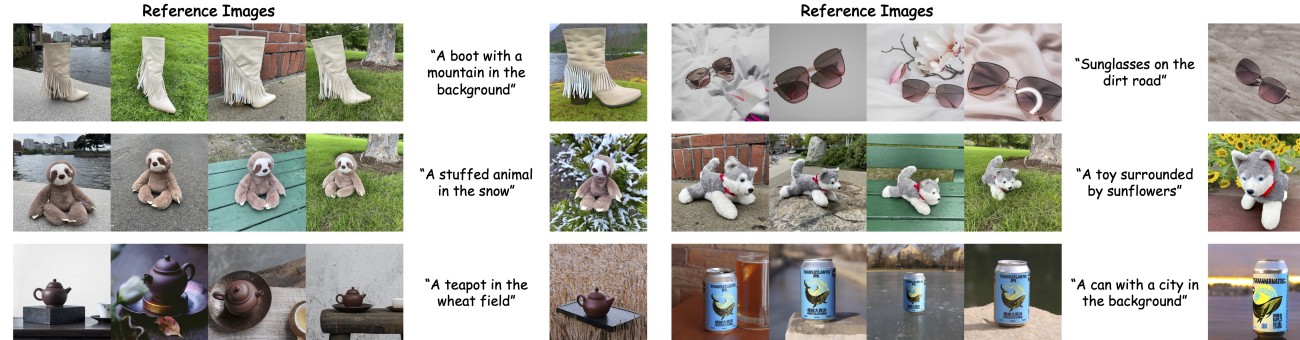

*Figure 8.* Visualization of generated samples on DreamBench.

*Table 3.* Evaluation comparisons on MRBench. "failed" means LoRA fails to generalize to the unseen held-out set.

| Method | CLIP-I ↑ | CLIP-T ↑ | DINO-I ↑ |
|---|---|---|---|
| *Held-in split* | | | |
| LoRA (Hu et al., 2021) | 0.831 | 0.715 | 0.654 |
| IP-Adapter-SDXL (Ye et al., 2023) | 0.768 | 0.632 | 0.527 |
| EasyRef | **0.856** | **0.739** | **0.675** |
| *Held-out split* | | | |
| LoRA (Hu et al., 2021) | failed | failed | failed |
| IP-Adapter-SDXL (Ye et al., 2023) | 0.813 | 0.646 | 0.611 |
| EasyRef | **0.838** | **0.715** | **0.653** |

*Table 4.* Quantitative comparisons on the DreamBench.

| Method | CLIP-I ↑ | CLIP-T ↑ | DINO-I ↑ |
|---|---|---|---|
| Real Images | 0.885 | - | 0.774 |
| *Tuning-based* | | | |
| Textual Inversion (Gal et al., 2022) | 0.780 | 0.255 | 0.569 |
| DreamBooth (Ruiz et al., 2023) | 0.803 | 0.305 | 0.668 |
| *Tuning-free* | | | |
| BLIP-Diffusion (Li et al., 2024a) | 0.779 | 0.300 | 0.594 |
| Re-Imagen (Chen et al., 2022) | 0.740 | 0.270 | 0.600 |
| IP-Adapter (Ye et al., 2023) | 0.793 | 0.330 | 0.612 |
| SSR-Encoder (Zhang et al., 2024) | 0.821 | 0.308 | 0.612 |
| $\lambda$-ECLIPSE (Patel et al., 2024) | 0.783 | 0.307 | 0.613 |
| Kosmos-G (Pan et al., 2023) | **0.822** | 0.250 | 0.618 |
| MoMA (Song et al., 2025) | 0.803 | **0.348** | 0.618 |
| EasyRef | 0.807 | 0.302 | **0.651** |

images that encompass various consistent elements. As presented in Figure 6, our method achieves better aesthetic quality and consistency with the original image prompts.

**Multi-image References.** We first compare our method with IP-Adapter and the tuning-based LoRA on the MR-Bench in Table 3. On the held-in split, the tuning-free EasyRef consistently achieves better performances than the tuning-based approach LoRA. In the zero-shot setting, the results demonstrate our method surpass the IP-Adapter with embedding averaging in alignment with the reference images and user prompt. We also present the qualitative visualizations in Figure 7.

Then we present the single-entity subject-driven generation performance comparisons on the DreamBench (Ruiz et al., 2023). We follow the original evaluation settings of DreamBooth. As presented in Table 4, EasyRef yields a comparable CLIP-I score to the tuning-based DreamBooth method. Moreover, it surpasses other tuning-free methods in DINO-I, which is the metric preferred by DreamBooth for evaluating subject preservation. We present the generated results in Figure 8. These experiments demonstrate our framework is capable of fully mining consistent visual elements among multiple reference images while maintaining strong generalization ability.

**Human Evaluation.** We systematically evaluate EasyRef

with IP-Adapter and LoRA in terms of reference consistency and aesthetic quality. The human evaluation is conducted on our proposed MRBench. Human evaluators were presented with pairwise image comparisons, one generated by EasyRef and the other by a competing model, under blind conditions to ensure fairness. As illustrated in Figure 9, EasyRef outperforms other models in both image-reference alignment and visual aesthetics in user study. This demonstrates EasyRef's capacity to generate high-fidelity images that conform to the provided reference images.

### 4.3. Ablation Study

**Scaling the Number of Reference Images.** Figure 10 illustrates EasyRef's performance across varying inference lengths. The model exhibits slightly robust performance across varying numbers of references when the number of reference images is within the training constraint. Specifically, the performances continue to increase as the number of references increases within the training constraint. However, performance degrades when the number of references exceeds this constraint. This is due to the limited number of groups with more than 16 images during training and the long-context finetuning may be inadequate. Moreover, the

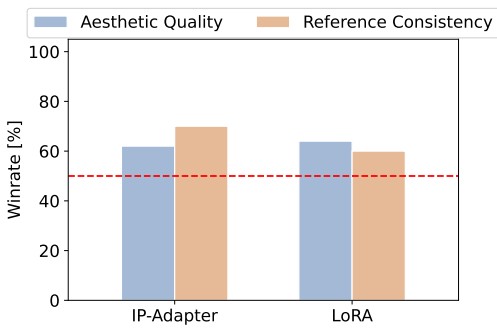

Figure 9. Comparisons of human preference evaluation on our MR-Bench. EasyRef can surpass other methods across the aesthetic quality and reference alignment.

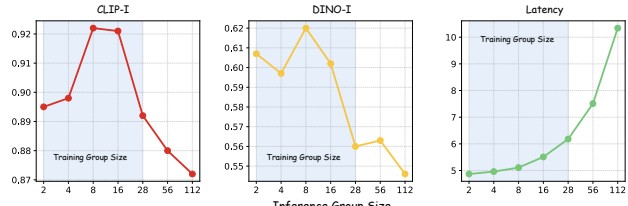

Figure 10. Evaluation of inference group size scaling. We randomly select 112 reference images and 1 target image-text pair with the same topic. "Latency" in the figure is measured in seconds per image.

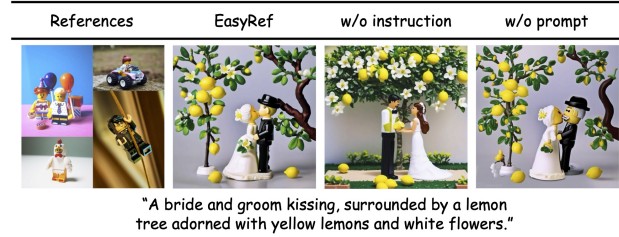

Figure 11. Impact of the multimodal instruction design.

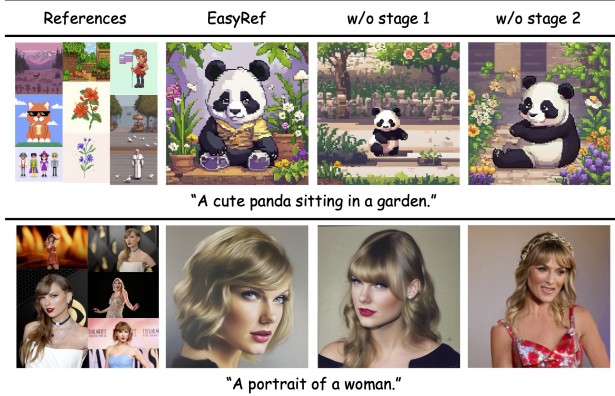

Figure 12. Effect of the progressive training scheme. "stage 1" and "stage 2" denote the alignment pretraining stage and single-reference finetuning stage, respectively.

inference efficiency of EasyRef is further evaluated and we find it still maintains acceptable efficiency with 56 reference images due to the effecient token aggregation design.

**Multimodal Instruction Input.** An ablation study was conducted to investigate the design of multimodal input to the LLM. As shown in Figure 11, the inclusion of instructions improves generation performance. The instruction leverages the MLLM's instruction-following ability to enable it to attend to desired contents within the reference images under explicit controls. Furthermore, incorporating the image prompt can exploit the text-understanding capacity of the MLLM and enhance text-image alignment.

**Progressive Training Scheme.** We visualize the impact of each stage on the model's ability to capture fine-grained visual details in Figure 12. For some reference contents, such as the pixel art style, EasyRef without alignment pretraining or single-reference finetuning maintains comparable performance. For reference images involving human facial identity preservation (*e.g.,* Taylor Swift), we find significant alignment improvements when adopting all training phases. We also investigate and visualize the effect of our training scheme in the Appendix A.2.

## 5. Conclusion

This paper presents EasyRef, a plug-and-play adaption method that empowers diffusion models to condition consistent visual elements (e.g., style and human facial identity, etc.) across multiple reference images under instruction controls. Our approach can effectively capture consistent visual elements within multiple reference images and the text prompt through an multi-image comprehension and instruction-following paradigm, while simultaneously maintaining strong generalization capabilities due to the integration of adapter-based injection. The proposed efficient reference aggregation strategy and progressive training scheme further enhance computational efficiency and fine-grained detail preservation. Through extensive evaluation on popular benchmarks and our newly introduced MRBench, EasyRef has demonstrably surpassed both tuning-free and tuning-based approaches, in terms of aesthetic quality and zero-shot generalization across diverse domains.

## Acknowledgements

This study was supported in part by the Centre for Perceptual and Interactive Intelligence (CPII) Ltd., a CUHK-led Inno-Centre under the InnoHK initiative of the Innovation and Technology Commission of the Hong Kong SAR Govern-

ment, and in part by NSFC-RGC Project N_CUHK498/24. Hongsheng Li is a PI of CPII under the InnoHK. This work was supported in part by the National Natural Science Foundation of China (Grant No. 62406141).

## Impact Statement

This paper presents work whose goal is to advance the field of Computer Vision and Machine Learning. There are many potential societal consequences of our work, none of which we feel must be specifically highlighted here.

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

# A. Appendix

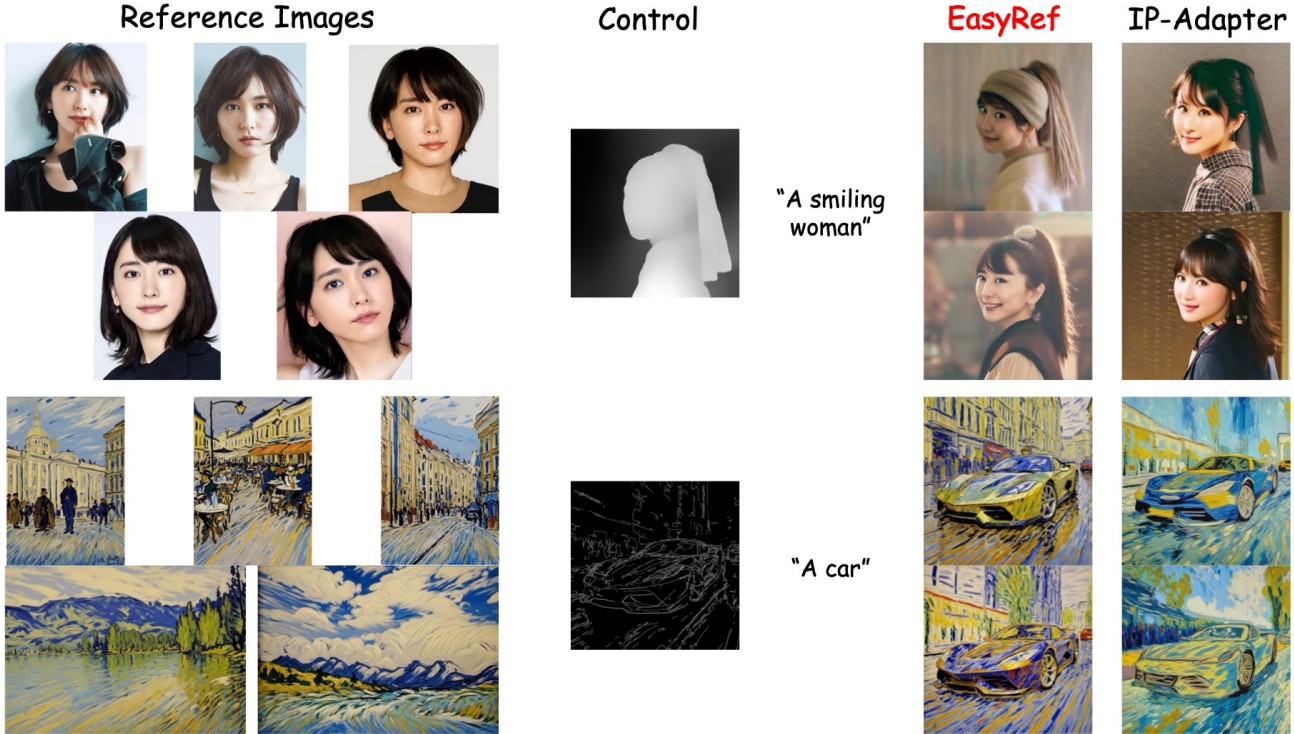

*Figure 13.* Comparison between EasyRef and IP-Adapter-SDXL with additional structure controls.

## A.1. Related Work

**Image Personalization.** Image customization in text-to-image diffusion model applications (Xue et al., 2024; Jiang et al., 2025; Shao et al., 2024b; Xue et al., 2025; Zhuo et al., 2024; Shen et al., 2025) aims to generate images that align with textual descriptions while incorporating specific visual attributes or references. Image personalization approaches can be categorized into tuning-free methods (Wang et al., 2024e; Zhang et al., 2023; Li et al., 2025; Jiang et al., 2024; He et al., 2024; Qi et al., 2024; Li et al., 2024c; Shi et al., 2024; Wang et al., 2024f; Patel et al., 2024; Zhang et al., 2024; Li et al., 2024b; Fu et al., 2023; Huang et al., 2024b) and tuning-based methods (Hu et al., 2021; Ruiz et al., 2023; Gal et al., 2022; Kumari et al., 2023). Tuning-free approaches typically extract visual representations, such as style and subject, from the reference image and inject these into the diffusion model. IP-Adapter (Ye et al., 2023) enhances image prompting capabilities through a decoupled cross-attention mechanism. Building upon IP-Adapter, InstantStyle (Wang et al., 2024a;b) injects CLIP (Radford et al., 2021) style embeddings into style-specific blocks. Both IP-Adapter-Face (Ye et al., 2023) and InstantID (Wang et al., 2024e) employ additional face encoders (Deng et al., 2019) to improve human facial identity preservation. A limitation of tuning-free methods is that they are trained with single-reference input, failing to fully exploit the consistent elements within multiple reference images. Tuning-based approaches, such as LoRA (Hu et al., 2021) and DreamBooth (Ruiz et al., 2023), finetuned the diffusion model using a limited set of images. Although tuning-based methods are capable of multi-image references, a key limitation is they necessitate specific finetuning for each distinct image group. In this work, we extend tuning-free methods to accommodate multiple reference images and the text prompt like tuning-based methods while maintaining robust generalization capabilities.

**Multimodal Large Language Models.** Multimodal large language models (MLLMs) (Liu et al., 2024a; Zong et al., 2024; Shao et al., 2024a; Chen et al., 2024; Wang et al., 2024c; 2025; Xiao et al., 2025b;a) have achieved impressive performance on open-world tasks, surpassing both traditional unimodal and multimodal approaches (Li et al., 2020; Zong et al., 2021; 2022; 2023a; Xue et al., 2022). Conventionally, MLLMs are constructed by integrating a pretrained large language model (LLM) (Lu et al., 2025; Wang et al., 2023; Lu et al., 2024b; Ren et al., 2024b; Lu et al., 2024a; Ren et al., 2024a; Zhou et al., 2023; Lu et al., 2024c) with encoders for additional modalities, such as vision. Pioneering works like LLaVA (Liu et al.,

*Table 5.* Ablation of reference token design.

| Number | Position | CLIP-I ↑ | CLIP-T ↑ | DINO-I ↑ |
|---|---|---|---|---|
| 32 tokens | -1 | 0.818 | 0.699 | 0.630 |
| 128 tokens | -1 | 0.832 | 0.711 | 0.650 |
| 64 tokens | -2 | 0.836 | 0.710 | 0.655 |
| 64 tokens | -3 | 0.833 | 0.708 | **0.656** |
| 64 tokens | -1 | **0.838** | **0.715** | 0.653 |

*Table 6.* Ablation of reference representation aggregation.

| Method | Average token number | CLIP-I ↑ | CLIP-T ↑ | DINO-I ↑ |
|---|---|---|---|---|
| Average | 144 | 0.825 | 0.694 | 0.619 |
| Concatenation | **797** | 0.832 | 0.700 | 0.624 |
| EasyRef | 64 | **0.838** | **0.715** | **0.653** |

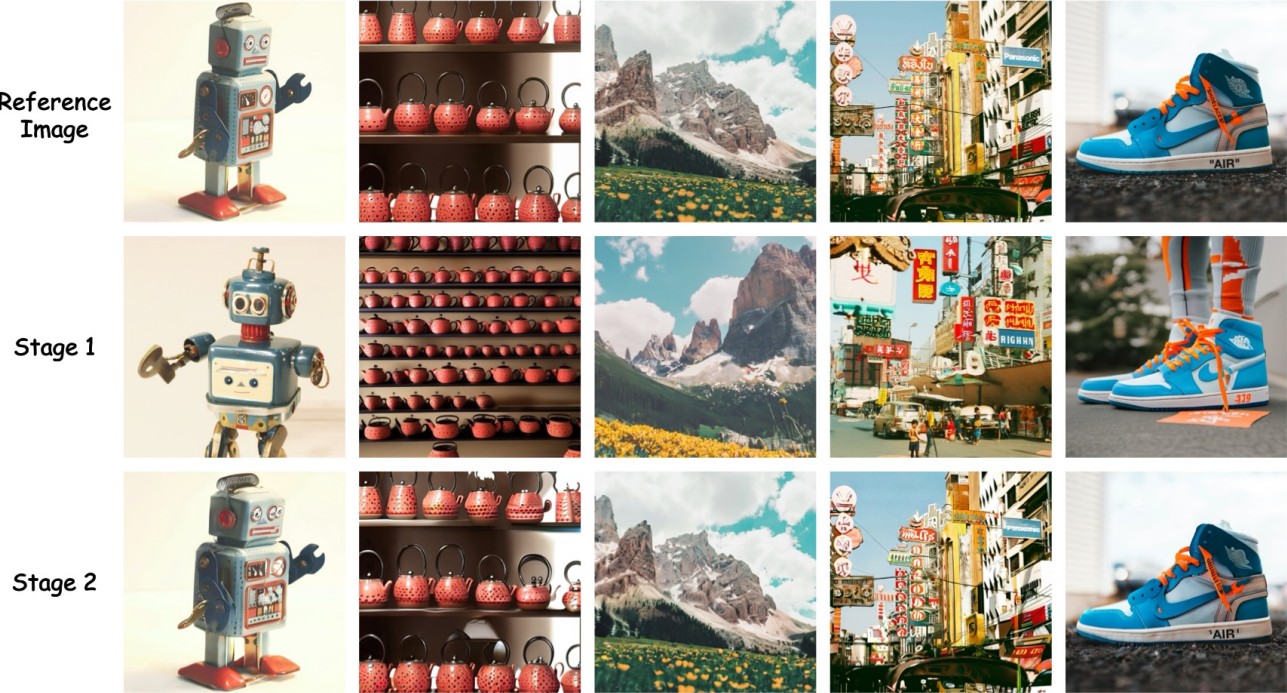

*Figure 14.* Single-reference generation visualizations. "Stage 1" and "Stage 2" refer to the alignment pretraining stage and single-reference finetuning stage, respectively.

2024b) and BLIP-2 (Li et al., 2023) consistently projected the vision representation from a pretrained CLIP vision encoder into the LLM for multimodal comprehension. Qwen-VL (Bai et al., 2023) collected massive multimodal tuning data and adopted elaborate training strategy for better optimization. The mixture-of-vision-experts designs, such as SPHINX (Lin et al., 2023), MoF (Tong et al., 2024), and MoVA (Zong et al., 2024), were explored to enhance the visual capabilities of MLLMs. Furthermore, models like LLaVA-NeXT (Liu et al., 2024a) and Qwen2-VL (Wang et al., 2024d) sought to enable the processing of images with arbitrary resolutions. LI-DiT (Ma et al., 2024) investigated how to effectively unleash the MLLM's prompt encoding capabilities for diffusion models. In this paper, we are the first to leverage the multi-image comprehension and instruction-following capabilities of the MLLM to jointly encode consistent representations of multiple reference images and the text prompt.

### A.2. More Experiments

**Compatibility with ControlNet.** As shown in Figure 13, our EasyRef is fully compatible with the popular controllable tool, ControlNet (Zhang et al., 2023). Compared to the IP-Adapter, EasyRef can generate high-fidelity, high-quality, and more consistent results when processing multiple reference images with additional structure controls.

**Reference Token Design.** We first ablate the number of reference tokens on the MRBench held-out split. The results in Table 5 show that too many or few tokens can hurt the performance. Hence, we choose 64 tokens to achieve the best trade-off between accuracy and efficiency. Furthermore, we observed comparable performances across various insertion

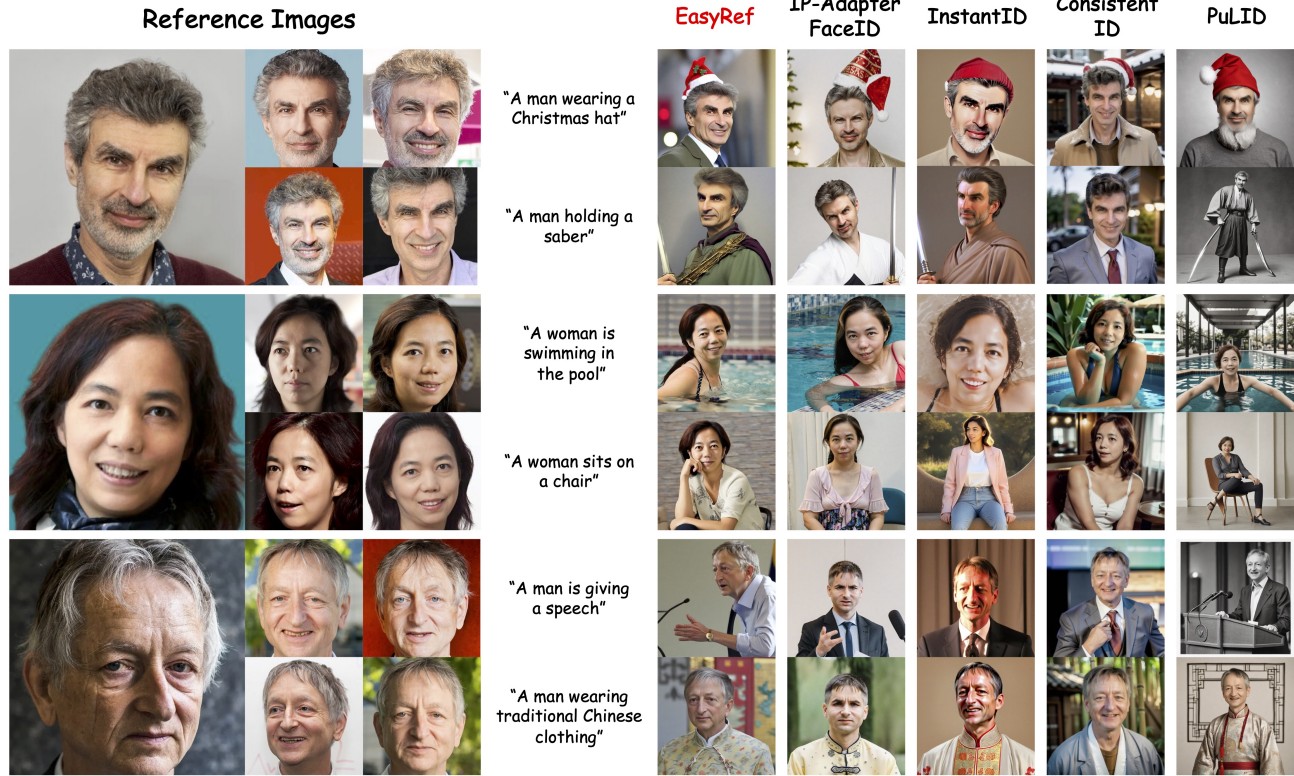

*Figure 15.* Qualitative comparison on human facial personalized generation.

positions (*e.g.*, the final, second to last, and third to last layers of the LLM) for the reference tokens. Consequently, we propose to insert the reference tokens into the final layer for optimal computational efficiency.

**Reference aggregation design.** In this experiment, we compare our reference token aggregation paradigm with embedding averaging and embedding concatenation. Specifically, we average or concatenate the vision encoder's representations of reference images. As shown in Table 6, averaging the multi-reference representations leads to performance degradation and the concatenation can increase the reference token number by more than $11\times$. Therefore, utilizing the multi-image comprehension and instruction-following capability of MLLMs can enhance the inference efficiency and performance.

**Fine-grained Detail Preservation.** Figure 14 illustrates how the progressive training scheme enhances the fine-grained detail extraction capabilities of MLLMs. Compared to the model trained solely in the first stage, the model further finetuned in the second stage demonstrates significantly improved preservation of fine-grained details, including logos, text, layouts, and other intricate elements. However, some text details of the reference image are not preserved due to the limited text rendering capability of the base Stable Diffusion XL.

**Human Facial Identity Preservation.** We also compare our EasyRef with other state-of-the-art specialist models (Ye et al., 2023; Wang et al., 2024e; Huang et al., 2024a; Guo et al., 2024) for human facial personalized generation in Figure 15. We observe that our method generally achieves high-quality generation, promising generation diversity, and strong facial identity fidelity.

### A.3. Preliminary

Denoising Diffusion Probabilistic Models (Ho et al., 2020) (DDPMs) are trained by maximizing the log-likelihood of the training data, given a data distribution $q(\mathbf{x}_0)$. The training process involves a forward diffusion process that gradually adds Gaussian noise to the data over $T$ timesteps:

$$q(\mathbf{x}_{1:T}|\mathbf{x}_0) = \prod_{t=1}^{T} q(\mathbf{x}_t|\mathbf{x}_{t-1}), \tag{7}$$

$$q(\mathbf{x}_t|\mathbf{x}_{t-1}) = \mathcal{N}(\mathbf{x}_t; \sqrt{\alpha_t}\mathbf{x}_{t-1}, (1-\alpha_t)\mathbf{I}). \tag{8}$$

Here, $\mathbf{x}_t$ represents the noisy data at timestep $t$ and $\alpha_t$ is a schedule parameter controlling the noise level at each timestep. The core of DDPM training lies in learning a parameterized model $p_\theta$ to approximate the reverse process:

$$p_\theta(\mathbf{x}_{0:T}) = p(\mathbf{x}_T) \prod_{t=1}^{T} p_\theta(\mathbf{x}_{t-1}|\mathbf{x}_t), \tag{9}$$

$$p_\theta(\mathbf{x}_{t-1}|\mathbf{x}_t) = \mathcal{N}(\mathbf{x}_{t-1}; \boldsymbol{\mu}_\theta(\mathbf{x}_t, t), \boldsymbol{\sigma}_t^2\mathbf{I}). \tag{10}$$

This model learns to progressively remove noise from a given noisy sample $\mathbf{x}_t$, recovering the original data $\mathbf{x}_0$.

Finally, with appropriate parameterization, the simplified per-timestep loss function becomes:

$$\mathcal{L}_t = \mathbb{E}_{q(\mathbf{x}_0, \mathbf{x}_t)} \left[ \frac{1}{2\sigma_t^2} \|\epsilon - \epsilon_\theta(\mathbf{x}_t, t)\|^2 \right], \tag{11}$$

where $\epsilon$ represents the Gaussian noise added during the forward process and $\epsilon_\theta(\mathbf{x}_t, t)$ is the model's prediction of this noise. Minimizing this loss function effectively trains the DDPM to denoise and generate high-quality samples.

