# OpenReview forum: "EasyRef: Omni-Generalized Group Image Reference for Diffusion Models via Multimodal LLM"
_ICML.cc/2025/Conference — ICML 2025 poster_

### Official Review · Reviewer_gzGG · 2025-03-10

**Overall Recommendation:** 2

**Summary:**

Leveraging the multi-image comprehension and instruction-following capabilities of the multimodal large language model (MLLM), this paper studies the personalization of diffusion models. It utilizes MLLM to capture the visual elements based on the provided reference images and instruction. The proposed framework could achieve style customization, character customization, and person id customization.

**Claims And Evidence:**

yes

**Essential References Not Discussed:**

This paper proposes to use MLLM to achieve controllable image generation. However, there are some methods [1,2] also using MLLM to achieve image editing. Although the tasks are different to some extent, both share the similar ideas. In addition, [3] using MLLM to achieve image generation should also be discussed.

[1] GUIDING INSTRUCTION-BASED IMAGE EDITING VIA MULTIMODAL LARGE LANGUAGE MODELS? ICLR'24

[2] SmartEdit: Exploring Complex Instruction-based Image Editing with Multimodal Large Language Models. CVPR'24

[3] UNIMO-G: Unified Image Generation through Multimodal Conditional Diffusion. ACL'24

**Experimental Designs Or Analyses:**

no issues.

**Methods And Evaluation Criteria:**

yes

**Other Comments Or Suggestions:**

-

**Other Strengths And Weaknesses:**

Strengths:

This paper builds a multi-reference image generation benchmark, which is helpful for the community.

The proposed method achieves better performance than previous works.

Weaknesses:

- The idea is not new.

In Tab.1, the paper makes comparisons between the proposed method and some previous methods. While Tab.1 shows the proposed method can support more functions, the difference between it and KOSMOS-G / MoMA regarding architecture and motivation is slight. For example, regarding the "multiple reference images", I believe such capability could be achieved by using a MLLM which supports the input of multiple reference images. The proposed method achieves better performance, which I believe makes sense, but the novelty is limited.

- In addition, there are some additional similar works not discussed [1,2,3].

[1] GUIDING INSTRUCTION-BASED IMAGE EDITING VIA MULTIMODAL LARGE LANGUAGE MODELS? ICLR'24

[2] SmartEdit: Exploring Complex Instruction-based Image Editing with Multimodal Large Language Models. CVPR'24

[3] UNIMO-G: Unified Image Generation through Multimodal Conditional Diffusion. ACL'24

**Questions For Authors:**

-

**Relation To Broader Scientific Literature:**

The proposed method and models could be used to achieve controllable image generation.

**Theoretical Claims:**

N/A

---

> ### Author Rebuttal · Authors · 2025-04-01
>
> Dear Reviewer gzGG,
>
> Thanks for your advice. We will address your concerns below.
>
> **Q1. Missing references and discussions.**
>
>    Thank you for raising this point. We have discussed the differences between these MLLM-based frameworks and EasyRef in the Q1 of rebuttal for Reviewer nUGh. We will cite and discuss these image editing and generation methods with MLLMs in the Related Work Section of our revised manuscript.
>
> **Q2. The novelty of this paper.**
>
>    Please refer to the Q1 of rebuttal for Reviewer nUGh.
>
> **Q3. Multi-reference consistent generation capability of other MLLM-based frameworks.**
>
>    1. Multi-image consistent generation is not achievable simply by supporting multi-image input in a framework; it requires architecture optimization, novel training algorithms, and collaborative integration with data construction to achieve the desired results.
>    2. Kosmos-G is another MLLM-based framework that can accept multi-image input. However, it focuses more on the composition of distinct elements and lacks the capability to extract consistent elements and generate consistent images based on those elements. For example, in its official setting for benchmarking DreamBench, which contains multiple images per instance, Kosmos-G uses only a single reference image. In addition, as shown in our comparison with EasyRef on the multi-image consistent generation task (https://github.com/anonymous-projectuser/image), visualizations demonstrate that EasyRef produces higher-fidelity and more consistent results.

---

> > ### Comment · Reviewer_gzGG · 2025-04-07
> >
> > Regarding [1] [2] mentioned in the review, the authors do not discuss these two works directly in the response. In the "Comparison with other methods" Part (Sec. 2.4), the authors mentioned "In contrast, we demonstrate that simply using reference tokens in the final layer of the MLLM can provide
> > sufficient reference information and efficiently inject conditions into the U-Net", while both [1] and [2] use learnable tokens in the mllm to guide the denoising process.
> >
> > I acknowledge the contribution in the training scheme (including data construction and optimization strategy), but the basic idea is similar to the previous works. Therefore, I maintain my initial score.
> >
> > [1] GUIDING INSTRUCTION-BASED IMAGE EDITING VIA MULTIMODAL LARGE LANGUAGE MODELS? ICLR'24
> >
> > [2] SmartEdit: Exploring Complex Instruction-based Image Editing with Multimodal Large Language Models. CVPR'24

---

> > > ### Author Response · Authors · 2025-04-07
> > >
> > > We thank the reviewer for the feedback. We would like to clarify the following points:
> > >
> > > **Q1** The idea is similar.
> > >
> > >    1. We acknowledge that MGIE [1], SmartEdit [2], and our work all utilize diffusion models and MLLMs with new tokens. However, the task addressed by EasyRef differs significantly from these frameworks. While both MGIE and SmartEdit focus on single-image editing with instruction comprehension, EasyRef is specifically designed for general multi-image consistent generation. To achieve this, we introduce several unique designs tailored for this task. The key differences are summarized in the table below:
> > >
> > > | Method | LLM image input | LLM linguistic input | LLM adapter | LLM distillation | Connector | Diffusion adapter |
> > > | - | - | - | - | - | - | - |
> > > | EasyRef | Multiple images | [reference instruction; user prompt] | inserting new tokens and enabling bidirectional attention in the final layer | N/A | MLP | Decoupled cross-attention layers |
> > > | MGIE [1] | Single image | [edit instruction; new tokens] | New tokens | Distill from Flan-T5-XXL (instruction loss) | Transformer | N/A |
> > > | SmartEdit [2] | Single image | [edit instruction; new tokens] | LoRA adapters, new tokens | Distill from CLIP text encoder (align loss) | Cross-attention layers | N/A |
> > >
> > >    2. MGIE and SmartEdit focus on using the MLLM's single-image and instruction comprehension for instruction-based image editing. In contrast, we leverage the MLLM's multi-image comprehension and prompt comprehension capabilities to understand multi-image contexts and user prompt, capturing consistent elements. Furthermore, our method enables explicit control of the reference encoding process through the MLLM's instruction-following ability.
> > >
> > > [1] GUIDING INSTRUCTION-BASED IMAGE EDITING VIA MULTIMODAL LARGE LANGUAGE MODELS? ICLR'24
> > >
> > > [2] SmartEdit: Exploring Complex Instruction-based Image Editing with Multimodal Large Language Models. CVPR'24

---

### Official Review · Reviewer_nUGh · 2025-03-14

**Overall Recommendation:** 4

**Summary:**

This paper proposes EasyRef, a plug-and-play method for diffusion models to generate consistent images from multiple references under instruction controls. It uses a multimodal large language model (MLLM) to capture consistent visual elements and introduces an efficient aggregation strategy and progressive training scheme to enhance performance and reduce computational costs. Additionally, a new benchmark (MRBench) is introduced for evaluating multi-reference consistent image generation. Experiments show that EasyRef outperforms existing methods in terms of consistency and generalization ability.

**Claims And Evidence:**

Yes, the claims made in the submission are supported by clear and convincing evidence.

**Essential References Not Discussed:**

N/A

**Experimental Designs Or Analyses:**

Yes.

**Methods And Evaluation Criteria:**

Yes.

**Other Comments Or Suggestions:**

Some figures in the paper, such as Figure 3, need significant refinement in terms of aesthetics. Issues such as oversized fonts are currently present.

**Other Strengths And Weaknesses:**

1. Strengths: The paper is well-organized and clearly motivated, focusing on addressing the limitations of existing methods in handling multiple reference images and fine-grained details. The proposed EasyRef framework and the Multi-Reference Generation Benchmark are both meaningful contributions. The experimental results are also promising.

2. Weaknesses: While it is reasonable to extract consistent visual elements from multiple reference images and text prompts via an MLLM, the EasyRef architecture lacks novelty. Additionally, the authors only discuss the performance gains compared to previous methods but do not analyze the increased computational complexity introduced by using an MLLM to process multiple reference images and text prompts.

**Questions For Authors:**

1. The visualizations presented by the authors currently focus on generating results from multiple reference images of the same single target. I am curious whether EasyRef can achieve customized generation when the multiple reference images contain the same two or more distinct targets.
2. The ablation studies in this paper seem somewhat incomplete. The authors does not explore the impact of different LLMs and different diffusion models on the proposed EasyRef framework.

**Relation To Broader Scientific Literature:**

The paper's contributions are closely related to the broader scientific literature on diffusion models and multimodal large language models (MLLMs). It builds on recent advancements in using MLLMs and diffusion models for image generation, addressing limitations of existing methods in handling multiple reference images and fine-grained details. The introduction of an efficient aggregation strategy and a new benchmark further aligns with ongoing efforts to enhance computational efficiency and standardize evaluation in this field.

**Theoretical Claims:**

Yes.

---

> ### Author Rebuttal · Authors · 2025-04-01
>
> Dear Reviewer nUGh,
>
> Thank you for appreciating our approach. We will address your concerns below.
>
> **Q1. The novelty of this paper.**
>
>    While we acknowledge that our project does not introduce groundbreaking new architectures, we wish to emphasize that it offers several important conclusions and methods that contribute novel insights to the community that are untouched by prior works. These conclusions and methods include identifying key limitations, developing effective data construction strategies, optimizing architecture and training for efficiency, and creating a new benchmark. Together, they provide a practical blueprint for building and evaluating a simple yet effective pipeline for multi-image consistent generation. We want to highlight three parts of differences.
>    1. Although some methods also utilize MLLMs and diffusion models, as acknowledged by Reviewer sCdX and nUGh, our design objectives, architecture and training optimizations, and tasks differ fundamentally. Other approaches leveraging MLLMs primarily use them for feature extraction in explicitly defined tasks (e.g., single-subject customization with a single reference image). In contrast, we demonstrate that MLLMs possess strong task instruction and multi-image comprehension capabilities, enabling them to extract consistent information based on instructions and multi-image contexts. Our method emphasizes the flexible understanding and extraction of consistent elements across multiple references, guided by instruction-based control.
>    2. We demonstrate that EasyRef with a simple architecture, can serve as a general-purpose model for multi-image customization and effectively cover various types of reference customization, such as subject, style, character, and face, rather than being limited to single-subject preservation or single-image subject-driven tasks. Consequently, EasyRef avoids the need for elaborate image preprocessing, such as subject-specific masking, making the process simpler and applicable to a broader range of scenarios.
>    3. By leveraging the unique capabilities of MLLMs combined with data construction methods and efficient designs, we achieve strong performance on the general multi-image consistent generation task. Our work focuses on building a comprehensive pipeline, including data construction, optimization of architecture and training strategies, and benchmark creation, rather than proposing an entirely new architecture.
>
>    Together, we believe these underscore the impact and utility of our work, providing practical insights and methodological advancements that benefit the broader research community.
>
> **Q2. Additional computational complexity introduced by using an MLLM to process multiple reference images and text prompts.**
>
>    We analyzed the model's performance and inference efficiency when varying the number of reference images, as shown in Figure 8. The table below presents the inference latency. The "baseline" refers to the standard SDXL text-to-image generation without image references. Our results show that incorporating an MLLM with multiple reference images only increases latency by 10%-20%.
>
> | Number of references | baseline | 1 | 2 | 4 | 8 |
> | - | - | - | - | - | - |
> | Latency | 4.31s | 4.78s | 4.87s | 4.96s | 5.11s |
>
> **Q3. Customized generation when the multiple reference images contain the same two or more distinct targets.**
>
>    We provide visualizations of customized generation using two subjects as references at https://github.com/anonymous-projectuser/image. As shown in the examples, the generated cat and dog exhibit high fidelity to their respective reference subjects. EasyRef leverages the instruction-following and multi-image comprehension capabilities of MLLM to flexibly identify and preserve consistent elements (e.g., two subjects) without the need for complex image preprocessing.
>
> **Q4. The impact of different LLMs and different diffusion models.**
>
>    We compare several EasyRef variants on MRBench, as shown in the table below. First, we observe that using a stronger base MLLM or diffusion model improves performance. Second, larger MLLMs significantly increase model complexity and latency. Therefore, we select Qwen2-VL-2B and SDXL to achieve the best trade-off between performance and efficiency.
>
> | MLLM | Diffusion Model | CLIP-I | CLIP-T | DINO-I | Latency |
> | - | - | - | - | - | - |
> | Qwen2-VL-2B | SD1.5 | 0.821 | 0.705 | 0.607 | 3.84s |
> | Qwen2-VL-7B | SDXL | 0.836 | 0.714 | 0.620 | 6.02s |
> | Qwen2-VL-2B | SDXL | 0.833 | 0.709 | 0.614 | 4.96s |

---

### Official Review · Reviewer_35bv · 2025-03-16

**Overall Recommendation:** 3

**Summary:**

This paper proposes to use a vision language model (VLM) to encode subjects in reference images, and convert them to soft tokens to personalize diffusion models. It can take objects, animals and human faces as subjects.

UPDATE after author response:

Per my request, the authors provided extra evaluation data on human face similarities compared with PuLID, InstantID and ConsistentID. It signals an impressive message that EasyRef outperform these SOTA face encoders. However, I'm unable to reproduce such results. Instead, my own runs of the EasyRef repo led to different conclusions: EasyRef performs poorly on human faces. I actually didn't require EasyRef to outperform SOTA methods on human faces, and just wished the authors to provide a reference data point. However, the authors didn't provide honest answers. In this regard, I decide to lower my rating to reject.

Update 2:
The authors provided sample images generated by their most recent model checkpoints. Seems the subject similarity indeed improves substantially compared to the (outdated) online demo. Therefore, I decide to revert my rating to a 3, as an appreciation of the tremendous efforts paid by the authors. Nonetheless, the compositionality/editability of the updated samples seems to be lower than the SOTA face encoders. Moreover, I still think the writing of this paper should be improved substantially, in particular, highlight novel architectural design choices and training recipies.

**Claims And Evidence:**

1. The claim that "Conventional methods encode consistent elements across reference images through averaging or concatenation (Shi et al., 2024),...  fail to capture desired visual elements through effective image interaction under explicit controls" seem to be inaccurate. The supporting example is "the IP-Adapter (Ye et al., 2023) generates an inconsistent image when the spatial locations of the target subject vary across the reference images" (Figure 2). What are the "Inconsistent Result" shown in figure 2? Different poses/views of the same person? Is this really inconsistent? Moreover, I don't think there's much issue with averaging or concatenation the subject embeddings. There's no spatial information encoded in subject embeddings, so the inconsistency across result images shouldn't be caused by averaging or concatenation.

**Essential References Not Discussed:**

A few SOTA face encoders are not mentioned, including PuLID and ConsistentID.

**Experimental Designs Or Analyses:**

1. Many baseline methods are too weak and non-SOTA. It's not necessary or useful to include them. For example in Fig 5., "SD image variations", "SD unCLIP", "Kandinsky 2.1", "Open unCLIP" etc.
2. When comparing with facial images, the SOTA methods are InstantID, consistentID, PuLID, etc. IP-Adapter FaceID is a weak baseline. The authors only compared with InstantID in the appendix. However, face similarities are not evaluated, which is one of the most important metrics on face embeddings.

**Methods And Evaluation Criteria:**

1. The method seems to be a straightforward scaling-up of MOMA (ECCV 2024). MOMA also uses a VLM (referred to as a "MLLM" in the MOMA and this paper) to encode subject characteristics and map them to subject embeddings. Moreover, MOMA also has a "Diffusion Learning Stage" that corresponds to the "Alignment pretraining stage". There are some minor differences on the architecture though, but I think the biggest difference is on the scale of the dataset (MOMA used 282K vs. EasyRef used 13M, 46x scaling up!)
2. Taking the drastically different data scales into consideration, it's not an apple-to-apple comparison to directly compare MOMA with EasyRef. An EasyRef model trained with similar data sizes would be more supportive of its architectural advantages, if there are any. Regardless of the data scale difference, EasyRef has almost identical performance as MOMA on DreamBench.

**Other Comments Or Suggestions:**

1. Although this paper follows the terminology of MOMA (ECCV 2024) to call the VLM as "multimodal LLM" (MLLM), personally I think it's more accurate to refer to it as "Vision language model" (VLM), because the LLM part of the model (in this paper, Qwen2-VL-2B) is quite weak compared with the commonly used MLLMs. Moreover, both MOMA and EasyRef don't use complicated language capabilities of the model. Therefore I think calling them VLMs would be more appropriate.
2. The first 3 pages are written disproportionately. The abstract and a cover image takes one page and the introduction takes another, followed by some diffusion equations which everyone in the field is familiar with. These parts can be greatly condensed. The most important part, Section 2.2 and 2.3, are written briefly. The saved space should be used for more technical details and discussions.

**Other Strengths And Weaknesses:**

Since EasyRef is just a simple and straightforward extension of MOMA (ECCV 2024) with 46x larger training data, I don't feel it has sufficient technical novelties to warrant publication in ICML.

**Questions For Authors:**

N/A

**Relation To Broader Scientific Literature:**

It's a simple and straightforward extension of MOMA (ECCV 2024). The biggest difference is it uses 46x larger training data, and encodes human faces in addition to objects and animals. There are some minor differences (e.g. adding a self-attention layer at the end of VLM to mix prompt tokens with the subject embeddings), but I don't think they are essential difference and don't have much impact on the overall performance.

**Theoretical Claims:**

N/A

---

> ### Author Rebuttal · Authors · 2025-04-01
>
> Dear Reviewer 35bv,
>
> Thanks for your comments. We will address your concerns below.
>
> **Q1. What are the "Inconsistent Result" shown in figure 2? There's no spatial information encoded in subject embeddings.**
>
>    1. The essence of the "Inconsistent Result" lies in the insufficient understanding of multiple reference features. Traditional methods fail to distinguish whether these features correspond to multiple subjects or different representations of the same subject. As a result, features corresponding to different spatial positions of the same subject are uniformly decoded, leading to the repetition of multiple subjects in the output.
>    2. Some subject-driven methods utilize subject-specific segmentation masks to exclude background and spatial information from the reference images. However, using a mask limits the generalizability of the framework. For instance, it becomes unsuitable for style and position-aware reference. General-purpose methods, such as IP-Adapter, avoid relying on masks, which can lead to such inconsistencies. Similarly, EasyRef does not elaborately preprocess input images, as it prioritizes generality over being limited to subject-driven tasks.
>    3. Additionally, using masks increases the complexity of the pipeline by requiring an extra segmentation model. Furthermore, inaccurate masks can degrade performance. For users, manually annotating masks during inference adds to the overall cost and reduces usability.
>    4. In cases where multiple subjects occlude each other (e.g., as discussed in Q5 of the rebuttal for Reviewer sCdX), using masks or bounding boxes may result in inaccurate subject embeddings.
>
> **Q2. The novelty of this paper.**
>
>    Please refer to the Q1 of rebuttal for Reviewer nUGh.
>
> **Q3. EasyRef uses significantly more training data compared to MOMA.**
>
>    1. MOMA is specifically designed for subject-driven generation task using a single reference image, whereas EasyRef is a general framework for multi-image consistent generation. EasyRef is capable of preserving various consistent elements, including subjects (e.g., common animals or objects), characters, styles, and human faces. Given their distinct design objectives and pipelines, directly comparing the data scales of the two approaches is neither fair nor meaningful.
>    2. Furthermore, our multi-image fine-tuning leverages approximately 350K clean subject-driven target images, which is comparable in scale to the data used by MOMA.
>
> **Q4. EasyRef has almost identical performance as MOMA on DreamBench.**
>    1. The DINO score is preferred by DreamBooth because its self-supervised training objective encourages the model to preserve fine-grained subject features. This makes it uniquely well-suited, compared to CLIP scores, for evaluating subject fidelity in generated images.
>    2. As shown in Table 3, EasyRef surpasses MOMA by 1.8% in DINO score. This performance gain is not trivial since MOMA improves the DINO score of IP-Adapter from 61.2% to 61.8% on the DreamBench.
>
> **Q5. It's not necessary to include baseline methods that are too weak and non-SOTA.**
>
>    We will carefully exclude some weak baselines in the final version.
>
> **Q6. Comparing EasyRef with SOTA facial reference methods and face similarities are not evaluated.**
>
>    We conducted qualitative comparisons (visualizations in https://github.com/anonymous-projectuser/image) showing EasyRef’s superior facial fidelity and quantitative evaluations using the face similarity metric adopted by PuLID on the MRBench:
>
> | Method       | Face Sim |
> | ------------ | -------- |
> | PuLID        | 0.632    |
> | ConsistentID | 0.652    |
> | InstantID    | 0.657    |
> | EasyRef      | 0.673    |
>
>    EasyRef outperforms SOTA methods in identity preservation, and these comparisons will be emphasized in the revised manuscript.
>
> **Q7. This paper follows the terminology of MOMA to call the VLM as multimodal LLM (MLLM), it's more accurate to refer to it as VLM.**
>
> We use "MLLM" to align with its growing adoption in the community for architectures combining vision encoders, projectors, and LLMs (e.g., LLaVA). While terms like "LMM", "VLM" or "LVLM" are also commonly used, our choice reflects current conventions rather than solely following MOMA.
>
> **Q8. The LLM of the model is quite weak.**
>
> Please refer to the Q4 of rebuttal for Reviewer nUGh.
>
> **Q9.  Both MOMA and EasyRef don't use complicated language capabilities.**
>
> Our synthetic captions (typically 1–3 sentences) are more detailed than conventional single-sentence prompts and require strong language comprehension capability to encode. Unlike MOMA, which only focuses on subject categories, EasyRef explicitly controls the extraction of consistent elements like face, style, and subject through instructions, better utilizing the model’s instruction-following capabilities.
>
> **Q10.  The first 3 pages are written disproportionately.**
>
> We will streamline the opening sections and expand Sections 2.2 and 2.3 in the revised manuscript.

---

> > ### Comment · Reviewer_35bv · 2025-04-06
> >
> > Thanks for providing extra evaluation data on human face similarities compared with PuLID, InstantID and ConsistentID.
> > It signals an impressive message that EasyRef outperform these SOTA face encoders. However, I'm unable to reproduce such results. Instead, my own runs of the EasyRef repo led to different conclusions: EasyRef performs poorly on human faces. I've tested 3 subjects with 1~3 reference images each, and generated 8 images (downloadable below):
> > https://limewire.com/d/2c7Il#hkVnlyEHhX
> > We can see that the face similarities are quite low on Trump and Fischoff, and are only good (comparable to SOTA methods) on Naran.
> >
> > I actually didn't require EasyRef to outperform SOTA methods on human faces, and just wished the authors to provide a reference data point. However, the authors didn't provide honest answers. In this regard, I decide to lower my rating to reject.
> >
> > Another negative point of EasyRef is that, it consumes a very high amount of GPU RAM. I got OOM on 48G GPUs. In order to run the demo, I had to pay by myself to use a 96G cloud GPU.

---

> > > ### Author Response · Authors · 2025-04-06
> > >
> > > We thank the reviewer for the feedback. We would like to clarify the following points:
> > >
> > > **Q1** Facial preservation performance is bad.
> > >
> > >    1. The model you evaluated was trained in November 2024, whereas the model described in this ICML submission was trained in January 2025 (before the ICML submission deadline). We withdrew our paper from CVPR 2025 and improved our model based on the feedback received from the CVPR 2025 comments.
> > >    2. When comparing with other SOTA facial preservation specialists, we follow their methodology [1][2], which employs an internal face detector to square-crop the human face in the reference image. This ensures that the face occupies the majority of the image. Example reference images can be found here: https://github.com/anonymous-projectuser/image/blob/main/face.png. This technique enhances generation results. However, we noticed that your reference images were not processed using this technique.
> > >    3. We observed that you only used a single reference image for the FisChoff case. However, our model is specifically designed for multi-reference consistent generation. As noted in line 265 of the paper, the minimum group size during finetuning is 2. Using only a single reference image may lead to suboptimal results from the network.
> > >    4. We conducted generation based on your tested reference images (Trump and Nashi) and prompts, and the results are shown in: https://github.com/anonymous-projectuser/image/tree/main/examples. Our results demonstrate good identity preservation capabilities.
> > >    5. Additionally, we provide visualization comparisons here: https://github.com/anonymous-projectuser/image/blob/main/face.png. All reference images and generated outputs are included. The results of other models were produced using their official checkpoints or demos hosted on their respective Hugging Face Spaces.
> > >
> > > **Q2** Inference OOM issue.
> > >
> > >    The Out of Memory (OOM) issue arises because 8 images are being generated per prompt. For comparison, generating 2 images requires 29 GB of memory, whereas the SDXL text-to-image pipeline uses 24 GB for the same generation setting. While our approach consumes slightly more memory, it introduces the capability for multi-image consistent generation.
> > >
> > > [1] InstantID: Zero-shot Identity-Preserving Generation in Seconds. (https://github.com/instantX-research/InstantID)
> > >
> > > [2] ConsistentID : Portrait Generation with Multimodal Fine-Grained Identity Preserving. (https://github.com/JackAILab/ConsistentID/blob/b42b725c49fcba83f57df63f9df610b703564447/pipline_StableDiffusionXL_ConsistentID.py#L74)
> > >
> > >
> > > **UPDATE:**
> > >
> > > Thank you for acknowledging our work and for raising the score.  We greatly appreciate your thoughtful review and constructive feedback. In our revised manuscript, we will substantially improve the paper's clarity through more polished writing, enhanced visual figures, and a refined experimental section that addresses all reviewers' concerns. We sincerely appreciate your time and effort in reviewing our paper. Thank you once again!

---

### Official Review · Reviewer_sCdX · 2025-03-17

**Overall Recommendation:** 4

**Summary:**

In the area of image personalization, tuning-free methods fail to capture consistent visual elements across multiple references, and tuning based methods require finetuning for new groups. In response, to learn a effective and efficient subject representation across a group of references, this paper proposes EasyRef, an approach that leverages MLLM's ability to capture the visual features across multiple images. In the representation learning process, a reference aggregation strategy and a progressive training scheme have been designed. To evaluate the performance, a paired image benchmark, MRBench, is collected.

**Claims And Evidence:**

- The paper claims that the proposed representation learning method can accurately extract the consistent details from multiple references and requires no optimization or test-time finetuning. While this claim is mostly correct (proved by Tab. 2 and 3), it is still questionable whether it can capture intricate details of subjects having complex textures (e.g., objects from DreamBench). Also the paired-data collection method (Sec. A.3) does not guarantee that only the same instance will be grouped together.
- The proposed method is very effective on capturing human faces and style, as demonstrated by the results in Fig. 5 and 6.

**Essential References Not Discussed:**

No.

**Experimental Designs Or Analyses:**

- (Minor) Fig. 2 shows the spatial misalignment issue of the embedding averaging, which provides valuable insights for similar works. However, this could be resolved by segmenting and cropping around the object. It would be more helpful if analysis/examples of other failure scenarios could be provided.
- EasyRef should be compared with more strong baselines in image customization, such as CustomDiffusion, MS-Diffusion, etc. Now the visual results are mostly about styles and human faces, and objects with complex textures (e.g., the rigid objects from DreamBench) are missing. More visual comparisons should be shown.
- The current pipeline of data clustering (Sec. A.3) is limited since it cannot ensure that only images of the same instance are grouped (e.g., images of the same category but different instance may also be included). This limitation will constrain the model's ability in detail preservation. Including object-centric video data may be one way to further improve identity preservation.

**Methods And Evaluation Criteria:**

- One of the main technical contributions, progressive training scheme is proved effective in Sec. 4.3; however, there should be more experiments ablating the design of the reference aggregation.
- In Tab. 3, the improvement of EasyRef seems marginal over the other baselines on DreamBench.

**Other Comments Or Suggestions:**

My concerns are mainly about the performance on identity preservation with complex objects; but the idea is solid and novel.

**Other Strengths And Weaknesses:**

- In Fig. 1, the task is not very clearly showed. It may lead to misunderstanding that the images on the right are generated by MLLMs. Maybe mentioned that Diffusion models are used for the customization.
- (Strength) In the second stage, the vision encoder of the MLLM is also trained, which is a reasonable setup to improve the quality and capacity of the learned representations.

**Questions For Authors:**

- During the alignment pretraining, are the MLLM and Diffusion trained together or is the MLLM the only model pretrained?

**Relation To Broader Scientific Literature:**

In image personalization, most previous methods focus on improving the diffusion model itself. However, in this paper, we leverage MLLMs as feature extractors and introduce a carefully designed representation learning stage to capture visual features, offering new insights to the community.

**Theoretical Claims:**

The equations (5-9) look reasonable and correct to me.

---

> ### Author Rebuttal · Authors · 2025-03-31
>
> Dear Reviewer sCdX,
>
> Thanks for appreciating our work and your advice. We will address your concerns below.
>
> **Q1. Can EasyRef capture intricate details?**
>
>    We provide visualizations on the DreamBench benchmark (https://github.com/anonymous-projectuser/image), demonstrating our method's ability to preserve intricate texture details. We also show a failure case in the last row. EasyRef fails to accurately preserve the textual details, primarily due to the inherent limitations in text rendering capabilities of the SDXL base model.
>
> **Q2. The collection method does not guarantee that only the same instance will be grouped together.**
>
>    1. We acknowledge that subjects belonging to the same category but not the same instance can be grouped together. However, this does not significantly impact subject-driven performance. Multi-image subject-driven generation requires that the main subjects across multiple reference images represent the same instance, and the generated outputs should contain the same instance. When multiple reference images depict subjects from the same category but not the same instance, using a target image with a subject from the same category is still a reasonable training approach.
>    2. To further enhance subject-driven performance, we have also constructed image groups using a stricter data filtering strategy (described in line 797 of our paper) by applying a higher DINO score threshold. Additionally, as discussed in line 250, we collected high-quality images and used them to train subject LoRA models. These LoRA models were then used for generating additional training data.
>    3. We randomly sampled 200 images from the training dataset and found that the proportion of such data is small (11 out of 200).
>
> **Q3. Ablating the design of the reference aggregation.**
>
>    We have already conducted ablation studies on the aggregation designs, as shown in Table 5 and Table 6 of our paper. Additionally, we provide more ablations in the table below:
>
> | Design | CLIP-I | CLIP-T | DINO-I |
> | ------ | ------ | ------ | ------ |
> | EasyRef | 0.833 | 0.709 | 0.614 |
> | + causal attention | 0.826 | 0.708 | 0.610 |
> | + inserting tokens before the first LLM layer | 0.836 | 0.710 | 0.615 |
> | + new cross-attention layers | 0.816 | 0.712 | 0.601 |
>
> From these results, we observe (1) using bidirectional attention in the final layer enables better interaction among reference tokens, (2) inserting reference tokens before the first layer of the LLM increases training costs but does not yield significant performance gains, and (3) the adopted decoupled cross-attention mechanism is more effective and efficient compared to adding new cross-attention layers.
>
> **Q4. The improvement of EasyRef seems marginal on DreamBench.**
>
>    Please refer to the Q4 of rebuttal for Reviewer 35bv.
>
> **Q5. It would be more helpful if other failure examples could be provided.**
>
>    We have included two additional failure cases for analysis in https://github.com/anonymous-projectuser/image:
>    1. Attribute Confusion: When the regions of the dog and the chair in the reference images significantly overlap (e.g., the dog partially or fully covers the chair), simply averaging the features from these reference images can result in attribute confusion. For instance, the generated dog might inherit the chair's color, while the chair adopts the dog's color, leading to inconsistent generation results.
>    2. Subject Hallucination: When the dog appears in front of the chair in one reference image but is seated on the chair in another, the simple fusion method may be misled by the positional discrepancy. This can result in subject hallucination, where the diffusion model generates an additional dog-shaped object on the chair.
>
> **Q6. EasyRef should be compared with more strong baselines.**
>
>    We will cite and discuss these state-of-the-art methods, such as CustomDiffusion [1], MS-Diffusion [2], and λ-ECLIPSE [3] in the final version. Their performance scores will be incorporated into Table 3.
>
> **Q7. In Fig. 1, the task is not very clearly showed.**
>
>    We will adjust the teaser figure and modify its caption to make the task clear.
>
> **Q8. During the alignment pretraining, are the MLLM and Diffusion trained together or is the MLLM the only model pretrained?**
>
>    We insert newly initialized reference tokens into the final layer of the LLM for aggregation. This makes the alignment pretraining process efficient, as we only train the final layer of the LLM, the reference tokens, the newly added cross-attention adapters in the U-Net, and the condition projector. The rest of the model remains frozen.
>
> [1] Multi-Concept Customization of Text-to-Image Diffusion
>
> [2] MS-Diffusion: Multi-subject Zero-shot Image Personalization with Layout Guidance
>
> [3] λ-ECLIPSE: Multi-Concept Personalized Text-to-Image Diffusion Models by Leveraging CLIP Latent Space

---

> > ### Comment · Reviewer_sCdX · 2025-04-09
> >
> > I appreciate all the additional experiments from the authors. My concerns have all been addressed and I have no more questions. I will maintain my current rating of accept.

---

### Decision · Program_Chairs · 2025-05-01

**Decision:**

Accept (poster)

**Comment:**

The paper reveived two Accepts, one Weak Accept, and one Weak Reject. Reviewers acknowledged the novelty/contribution of this paper, the contribution of the benchmark dataset, and promising experimental results. Especially, Reviewer 35bv run the demo code of the proposed EasyRef method and confirmed its effectiveness. Negative points majorly raised for the discussion of novelty, while authors provided detailed rebuttal and all reviewers acknowledged that their concerns had been addressed except that Reviewer gzGG was still not very satisfactory with it but was not against acceptance of the paper. Reviewer gzGG mentioned three papers which used MLLM for image editing. Authors listed detailed differences of their mehtod compared to these three existing methods. While the tasks are different (image generation vs. image editing), the compared difference seems reasonable. Therefore, considering that all reviewers acknowledged the contribution of the paper and accepted the rebuttal, except the last concern which was almost addressed, AC is happy to accept the paper for publication. Authors are required to update the rebuttal and discussion contents to the camera-ready version of the paper to improve it so as to address the raised concerns in the final paper.